# A socio-ecological approach to understanding the factors influencing the uptake of intermittent preventive treatment of malaria in pregnancy (IPTp) in South-Western Nigeria

Gertrude N. Nyaaba[1]*, Atinuke O. Olaleye[2]*, Mary O. Obiyan[3], Oladapo Walker[4], Dilly O. C. Anumba[1]

1 Academic Unit of Reproductive and Developmental Medicine-Obstetrics and Gynaecology, Faculty of Medicine Dentistry and Health, The University of Sheffield, Sheffield, United Kingdom, 2 Department of Obstetrics and Gynaecology, Benjamin Carson (Snr) School of Medicine, Babcock University, Ilishan, Nigeria, 3 Department of Demography and Social Statistics, Faculty of Social Sciences, Obafemi Awolowo University, Ile-Ife, Nigeria, 4 Department of Pharmacology, Benjamin Carson (Snr) School of Medicine, Babcock University, Ilishan, Nigeria

* g.nyaaba@sheffield.ac.uk (GNN); olaleyea@babcock.edu.ng (AOO)

**Data Availability Statement:** Data cannot be shared publicly because of participant identifying details which does not ensure confidentiality of

## Abstract

Malaria in pregnancy (MiP) remains a key cause of poor maternal and neonatal health outcomes, particularly in the African region. Two strategies globally promoted to address MiP require pregnant women in malaria-endemic regions to sleep under insecticide-treated bed nets (ITNs) and take at least three doses of intermittent preventive treatment (IPTp) during pregnancy. Yet, several multilevel factors influence the effective uptake of these strategies. This study explored the factors for the poor uptake of IPTp and use of ITNs in lower socio-economic communities in Nigeria. We conducted semi-structured interviews (SSI) and focus group discussions (FGD) with a total of 201 key stakeholders in six communities in Ogun State, South-Western Nigeria. Twelve SSIs were conducted with traditional birth attendants (TBAs), faith-based birth attendants and healthcare providers operating in public health facilities. Community leaders (7), pregnant women (30) and 20 caregivers were individually interviewed. Sixteen FGDs were conducted with multi- and first-time pregnant women grouped by location and pregnancy experiences. A thematic approach was used for data analysis. At the individual and social levels, there is a high general awareness of MiP, its consequences and ITNs but low awareness of IPTp, with type of antenatal care (ANC) provider being a key factor influencing access to IPTp. The choice of ANC provider, which facilitates access to IPTp and ITNs, is influenced by the experiences of women, relatives and friends, as well as the attitudes of ANC providers and community perceptions of the type of ANC providers. Concurrent use of multiple ANC providers and ANC providers' relationships further influence acceptability and coverage for IPTp and ITN use. At the health sector level, there is low awareness about preventive malarial strategies including IPTp among TBAs and faith-based birth attendants, in contrast to high IPTp awareness among public healthcare providers. The findings highlight several factors that influence the utilisation of IPTp services and call for greater synergy and collaboration between the three

study participants. Data are available from the Ethics Review board of Babcock University (contact via olaleyea@babcock.edu.ng) for researchers who meet the criteria for access to confidential data.

**Funding:** This manuscript is part of a larger study funded by a QR-GCRF (Research England, University of Sheffield). Sustainable partnerships Grant X/159770 awarded to Professor Dilly OC Anumba. The funders did not play any role in the design of the study, data collection, analysis and development of this manuscript.

**Competing interests:** The authors have declared that no competing interests exist.

groups of healthcare providers towards enhancing access to and acceptability of IPTp for improving maternal and child outcomes.

## Introduction

Malaria prevention remains a key public health challenge, with malaria in pregnancy (MiP) affecting an estimated 11 million pregnancies in sub-Saharan Africa (SSA) [1]. About 93% of the estimated 405 000 global malaria deaths in 2018 occurred in the African region, with malaria reported as the third commonest cause of death among women of reproductive age in SSA [1–3].

The World Health Organisation (WHO) recommends that in addition to sleeping under insecticide-treated nets (ITNs), starting from the second trimester of pregnancy, pregnant women in endemic regions should take at least three doses of intermittent preventive treatment in pregnancy (IPTp) to prevent malaria [2]. Evidence shows that IPTp with sulfadoxine-pyrimethamine (SP) reduces maternal anaemia, low birth weight, and perinatal mortality [2]. Yet, coverage and utilization of IPTp remain sub-optimal in SSA where about 40% of pregnant women did not sleep under an ITN and over 60% did not receive the recommended three or more doses of IPTp in 2018 [1–3]. Operating within the context of weak health systems grappling with the heaviest burden of malaria and the highest global maternal and perinatal mortality rates, uptake of IPTp and use of ITNs is crucial for improving maternal and perinatal health outcomes.

Nigeria, the most populous SSA country, contributed 25% of the estimated global malaria deaths in 2018 despite adopting the policy of free distribution of ITNs (2001) and IPTp (2004) as MiP intervention strategies [1, 2]. Under these policy directives, pregnant women freely receive ITNs and IPTp at antenatal clinics (ANC) in public health facilities and facilities managed by non-governmental organisations [4]. Yet, in 2018, only about 20% of pregnant Nigerian women attending ANC received at least three doses of IPTp-SP [1]. The Nigerian Demographic and Health Survey 2018 (NDHS) shows that 58% of pregnant women had slept under an ITN the preceding night and while about 57% of women attended four or more ANC visits, only 17% of women with a live birth in the last two years had received at least three IPTp doses [5]. A few studies have shown some health system and individual factors [4, 6] as key contributors to the low uptake of ITN and IPTp in Nigeria. However, there is scant evidence on how interpersonal, societal and provider related factors contribute to the poor uptake of ITNs and IPTp, particularly in rural and lower socio-economic communities where social and communal norms are most influential. This is relevant as Nigeria operates a pluralistic health sector with about 60% of Nigerians seeking private health care [7, 8]. Traditional Birth Attendants (TBAs) and faith-based birth attendants, as key parts of the private health sector, provide a significant amount of ANC services [9, 10], especially in rural and lower socio-economic communities. These peculiarities have crucial implications for access to and utilisation of ITNs and IPTp within such contexts.

Using elements of the socio-ecological model (SEM), this study explored the factors contributing to the poor coverage of ITN and IPTp within a pluralistic health sector in Ogun State, southwest Nigeria. As a theoretical framework, the SEM recognises that peoples' health behaviours are embedded within larger interactive and overlapping social systems which influence health outcomes [11]. It thus presents an apt framework for exploring the intricate multi-layered factors within the social system that shape pregnant women's acceptance and utilisation of interventions for the prevention of MiP.

## Materials and methods

### Study design

This was a multi-site cross-sectional qualitative study, using semi-structured interviews (SSI) and focus group discussions (FGDs) to collect primary data from 201 participants in six semi-urban and rural communities in Ogun state between February and March 2019. Initial study questions which guided data collection were developed using grounded theory [12, 13] as part of a larger study on determinants of malaria prevention during pregnancy in Ogun state, Nigeria. Conducted following the consolidated criteria for reporting qualitative research [14] (S1 File), all data was collected as digitally recorded audio files by 14 Research Assistants (RAs) with at least two years contextual research experience. The authors developed the topic guides (S2 File) for data collection based on existing literature on MiP interventions in Africa and Nigeria specifically and drawing on their own conceptual experiences working on maternal and child health issues in the study areas. Interviews and discussions were conducted in private locations, usually identified by the participant (s) at his/her convenience on a scheduled date after verbal and written/thumb printed consent was sought from each participant. Ethical approval for the study was received from the Ethics Review boards of Babcock University (BUHREC056/19) and the Ogun State Ministry of Health (HPRS/381/290). The research team had no prior contact with study participants.

### Study area

The study was conducted in three semi-urban and rural communities within Ogun State, southwest Nigeria. The state borders Lagos State to the south, Oyo and Osun states to the north, Ondo to the east and the Republic of Benin to the west. Abeokuta is the capital and largest city in the state. Ogun state occupies an area of 16,981 km$^2$ and is divided into three geopolitical zones (East, West and Central), with 20 administrative units known as local government areas (LGAs); each LGA consists of several administrative wards. Three LGAs (Ijebu-Ode, Sango-Ota, Odeda) were randomly selected from each geopolitical zone after which one semi-urban and one rural community was purposively selected from each LGA. From Ijebu-Ode LGA, Porogun was selected as the semi-urban community and Itamapako as the rural community. Sango was selected as the semi-urban community and Ketu as the rural community in Sango-Ota LGA. In Odeda LGA, the semi-urban community selected is Obantoko, while the rural community is Odeda.

The urban areas consist of many public and private health institutions, with many of the community members engaged in the formal sector. There is primary, secondary, and tertiary level of healthcare services provided within the semi-urban communities. However, traditional birthing homes which are often managed by traditional birth attendants (TBAs) also exist. In contrast, rural areas consist of mostly petty-traders and farmers. A few government offices are situated in rural areas with limited primary and secondary schools available. Only primary healthcare facilities are often available within rural communities, usually managed by a nurse and mid-wife. There are many TBAs in rural areas with high patronage from the community.

### Methods

SSIs were used to explore in-depth, perceptions and experiences regarding uptake of ITN and IPTp in six rural and semi-urban communities in Ogun West, Central and East districts. We purposively selected public healthcare providers (4), TBAs (4) and faith-based birth attendants (4) based on popularity and accessibility within selected communities, to explore access and awareness surrounding IPTp in a pluralistic health sector. We further purposively contacted

community leaders (7) and caregivers, mainly, family members (20), through participating pregnant women (30), to enable us to explore the influence of community and family networks on IPTp access and uptake. A sample size of between 20 to 30 interviews in qualitative studies permits data saturation where the key themes are addressed and additional interviews do not add new themes [14–16] and so a predetermined sample size of 69 SSIs was considered sufficient to reach data saturation. SSIs were conducted in the local language (Yoruba) and/or English with each interview ranging from 30 to 65 minutes.

Sixteen FGDs were conducted with 132 pregnant women attending ANC, grouped by location and ANC provider (TBA, faith-based birth attendants and public healthcare providers) to gain insights into their collective pregnancy experiences with malaria prevention strategies. RAs recruited seven to eleven pregnant women at ANCs for FGDs with discussion time ranging between 40 to 76 minutes. Study information was given verbally and information sheets clearly outlining the purpose of the research presented to participants. Two trained RAs moderated the discussions in the local language using a topic guide and took notes.

## Data analysis

Because of the iterative nature of the study, data collection, translation, transcription and analysis were concurrent to enable the exploration of emerging themes. Six native speakers transcribed digitally recorded interviews and discussions into Yoruba, which were then translated into English. Co-authors reviewed all transcripts and translations to ensure the quality and accuracy of the translation. Transcripts were not returned for participant crosschecking due to low literacy rates. A thematic approach was used in data analysis with the initial coding framework generated in QRS Nvivo 11 pro by the first three authors using pre-identified themes derived from the initial topic guide with emerging themes. Transcripts were coded in constant comparison and codes reviewed for contextual relevance. Patterns and linkages between quotes, codes, themes and existing literature were explored in-depth to identify areas of convergence and divergence.

## Results

Table 1 below provides the background characteristics of all study participants.

### Malaria in pregnancy

**Perceptions regarding MiP.**   MiP was perceived as common during pregnancy, with adverse outcomes such as miscarriages, premature births, impaired foetal growth and development and neonatal deaths. A few pregnant women attending TBA clinics held the view that pregnant women do not get malaria. On the consequences of untreated MiP, both public and private healthcare providers emphasised delivery complications while pregnant women and caregivers emphasised the fear of new-borns developing measles, typhoid fever, jaundice and convulsions. They related that malaria caused stomach pains which resulted in miscarriages and preterm births. They also related that malaria could cause weakness in mothers and convulsions in newborns, or hinder the baby's physical growth. The personal experiences of women, and those of family members and friends, of malaria, its prevention and treatment were commonly reported by pregnant women, caregivers and community leaders as the key sources of information for preventing and treating MiP.

*"It can make someone give birth to premature baby. . .the baby might die. . .it can spoil the pregnancy. . ..it can abort the pregnancy. . .it can cause measles on the baby. . . if the pregnant woman has malaria it can let the baby fall [miscarriage] because her stomach will be paining*

**Table 1. Characteristics of study participants (N = 201).**

| 1A. Characteristics of pregnant women participating in FGDs conducted in Yoruba (N = 132) | | | | | | | |
|---|---|---|---|---|---|---|---|
| Location | Type of ANC attended | Quotation ID | Number of FGDs conducted | Number of participants | Age range | Number of children | Education |
| Rural location | Primary health center | FGD-PW-PHC-R | 2 | 8–10 | 18–39 | 0–4 | None-Secondary |
| | TBA center | FGD-PW-TBA-R | 2 | 8–10 | 18–40 | 0–4 | None-Secondary |
| | Faith-based center | FGD-PW-FBF-R | 1 | 7 | 19–28 | 0–4 | None-Secondary |
| Semi-urban location | Primary health center | FGD-PW-PHC-S | 3 | 8–11 | 18–39 | 0–4 | None-tertiary |
| | TBA center | FGD-PW-TBA-S | 1 | 10 | 18–36 | 0–4 | None-primary |
| | Faith-based center | FGD-PW-FBF-S | 1 | 8 | 18–30 | 0–4 | None-tertiary |
| Urban location | Primary health center | FGD-PW-PHC-U | 3 | 7–9 | 18–40 | 0–4 | None-tertiary |
| | Faith-based center | FGD-PW-FBF-U | 2 | 7–10 | 20–32 | 0–4 | None-secondary |
| 1 B: Characteristics of semi-structured interviews participants (N = 69) | | | | | | | |
| Type of Participant | Location | Quotation ID | Number of SSIs conducted | Educational Status | | Language | |
| Pregnant women | PHC, TBA and faith-based birthing homes | IDI-PW-(PHC/TBA/FBF) | 30 | Primary—tertiary | | Yoruba | |
| Caregivers/family members | Rural and semi urban Ogun | IDI-CG-(PHC/TBA/FBF) | 20 | Primary—tertiary | | Yoruba/English | |
| Community leaders | Rural and semi-urban Ogun | SSI-CM- | 7 | Primary—secondary | | Yoruba/English | |
| TBAs | Rural and semi urban Ogun | SSI-TBA- | 4 | None -primary | | Yoruba | |
| Faith-based birthing homes | Private birthing homes | SSI-FBF- | 4 | Secondary—tertiary | | Yoruba | |
| Healthcare providers | PHC, CHO | SSI-PHC-R/S/U (Rural/Semi-urban/Urban) | 4 | Tertiary | | English | |

her and she might give birth early . . .malaria won't allow the baby to move in the stomach
. . .it can cause convulsion for the baby, the baby might not talk or sit on time, and it won't cry
and until they beat the baby before it will cry. . .the mother might not have the strength to
push during labour."

*FGD-PW-PHC-R -01*

"It happened to my sister, she lost a baby last year. . . it weakens the mother, hinders the
growth of the baby. . . some children might have red or yellow eyes. . . it can make it difficult
for the mother to deliver easily. . . if too severe, it can damage the baby's body parts, like hand-
icapped kids."

*FGD-PW-PHC-U-02*

**Practices for preventing MiP.** Pregnant women, caregivers and TBAs commonly
reported the practice of personal, home and environmental hygiene and the use of traditional
remedies consisting of boiled leaves of pawpaw, mango and ginger as key strategies to prevent-
ing MiP. ITN use was the most commonly reported method for preventing MiP among all

participants, with only public healthcare providers and a few pregnant women reporting IPTp as a MiP prevention strategy. A few pregnant women attending ANC at public health facilities reported receiving medicines for preventing MiP although the majority did not know the name of the medications. IPTp was less commonly reported by faith-based birth attendants and was not reported by any TBA. Some pregnant women and caregivers held the view that taking medications during pregnancy was good for the health of mother and baby while several other pregnant women, caregivers and TBAs raised concerns about taking medications during pregnancy in the absence of illness.

*"Avoid mosquito. . . malaria herbs. . . sleep under net. . . pregnant woman shouldn't be walking under sun. . .clean the environment. . .. take fansidar [SP]. . . . . . if someone should use unripe pawpaw . . . peel it, put the leaf and wash it, then put it inside water. . . they use it to bath. . . I did it with my children and they are grown up and can stand very well because of things [remedies] I used for them during their pregnancy that is working in their body"*

*FGD-PW-PHC-U-01*

*"Most times some of those drugs can be overactive in the body so one has to be careful. . . I have not used drugs to prevent malaria"*

*FGD-PW-PHC-S-02*

**Access to and utilisation of ITNs.** All participants commonly reported ITNs as freely accessible from public health facilities. Most rural dwelling pregnant women, caregivers and community leaders additionally reported periodic distribution of free ITNs by non-governmental organisations (NGOs). Pregnant women attending TBA and faith-based ANC did not report receiving ITNs from their providers, with several pregnant women reporting that they accessed ITNs freely from public health facilities, NGOs, or bought ITNs from commercial sources. Only a few pregnant women who owned ITNs reported sleeping under ITNs.

*"I also protect myself through the use of net every night. . . they gave a net me at the hospital and my mother gave me one."*

*SSI-PW-PHC-S-01*

*"They didn't give us net here. You have to go to the clinic to get it."*

*SSI-PW-TBA-S-02*

*"We've given them mosquito nets to prevent them from malaria. . .They are given nets the same day they get their [registration] cards, and we orientate them how to make use of the nets."*

*SSI-PHC-S-01*

**Access to and utilisation of IPTp.** Nearly all participants reported that pregnant women receive medications when they attend ANC clinics, although only a few pregnant women specifically mentioned SP or its common brands (e.g. fansidar, amalar) as a preventive antimalarial. Medications like folic acid were commonly reported by pregnant women and a few caregivers as provided at public ANC clinics but these are not provided by TBAs or faith-based birth attendants. Some pregnant women attending TBA clinics reported being encouraged by TBAs to buy antimalarial medications from commercial sources when TBAs suspected they

had MiP. Only a few pregnant women attending public ANC facilities mentioned ever receiving or being told they would receive IPTp three times during pregnancy. Other pregnant women reported receiving injections to prevent MiP although they did not know the name of the injection. Even fewer pregnant women reported taking IPTp as a directly observed therapy. Caregivers and community leaders reported that they encouraged pregnant women to take their medications when they were home, although no caregiver specifically mentioned SP for preventing MiP. No TBA and only a few faith-based birth attendants mentioned IPTp for preventing MiP. Nearly all ANC providers at public health facilities were aware of IPTp as a preventive strategy for MiP and reported providing IPTp during ANC services, with a few reporting the use of SP as treatment of MiP.

> *"There is a drug given to us when we go to the clinic to prevent malaria. I don't know the name but it's yellow. I think it's called "amalar" and we take it home to use. . . They give us drugs like "Amalar", "folic acid" to prevent malaria at home."*
>
> *FGD-PW-TBA-S-02*

> *"We give them IPT when their pregnancy is 16 weeks, we give them at 20 weeks, and at 24 weeks. Before it was twice, but now I think it is 5 times—every four weeks, once the pregnancy is 20 weeks or 16 weeks. The IPT has a particular brand name called SP. I think there was a national guideline to prevent malaria."*
>
> *SSI-PHC-U-01*

### Factors influencing IPTp uptake and use of ITNs

**Side effects of ITN use and IPTp uptake.** Pregnant women and caregivers commonly reported side effects such as heat from the chemicals used in treating the ITNs, developing rashes and itching as reasons for pregnant women's non-use of ITNs. Difficulties in swallowing, nausea and other side effects such as sweating, drowsiness and frequent urination were commonly reported by pregnant women as key reasons for their poor uptake of IPTp.

> *"I do feel nauseated when I take that drug at the hospital. . .When I use the drug, I feel weak, when I now use it, I will just sleep. . . If I take it with water, I will vomit it."*
>
> *FGD-PW-TBA-R-01*

> *"When I take the drug it changes the movement of the child. . .I feel like the baby is shifting towards my anus"*
>
> FGD-PW-PHC-U-03

> *"It [ITNs] usually causes heat. And when I first tried it then it gave me rashes, they now said because I didn't dry it out. . . then I didn't even like it anymore."*
>
> *FGD-PW-TBA-R -01*

**Inadequate health education on ITN use and IPTp.** Pregnant women reported a lack of education on ITN use and IPTp from their ANC providers. Some pregnant women indicated that they were supposed to use the ITNs after they had delivered their babies. Others reported that they did not receive any information on when and how to use the ITNs so they did not

frequently use them. Several pregnant women reported that they did not receive any information about IPTp from their ANC providers. Some pregnant women added that the combination medications they take routinely in pregnancy are supposed to be sufficient for MiP prevention and so when they are given drugs at the ANC to take at home, they did not take them. Other pregnant women reported that they had difficulties swallowing drugs and hence indicated a preference for injections, as they often did not swallow the drugs given to them at the ANC. Most caregivers added that their wards received check-ups and medications to take at home from ANC providers but could not tell the exact reason why they were to take the drugs and often reported that it was to ensure the health of the baby.

> *"I have not used it [ITN]. I thought we will start using it after giving birth."*
>
> *SSI-PW-FBF-R-02*

> *"I use pregnant care [vitamin supplements] so I'll collect it [SP] and leave it at home. . . they don't say anything about the drugs so all the drugs given to me is already in pregnant care which is 2in1. . ."*
>
> *FGD-PW-TBA-S-01*

> *"They brought it [ITNs] but there was no lecture"*
>
> *SSI-PW-PHC-R-01*

**Financial barriers.** Several pregnant women, caregivers and community leaders reported financial constraints in procuring prescribed medications including SP and ITNs. This was reported as a result of shortages of ITNs and SP at public health facilities where they are freely provided. The financial difficulties included transportation costs resulting from geographical barriers to public ANC facilities where IPTp and ITNs could be freely accessed. This was confirmed by ANC providers in public health facilities who elaborated that shortage of freely provided SP and ITNs in their facilities required that they write out prescriptions for pregnant women to buy them in commercial shops. They added that this was not ideal as they were unable to observe the women taking the medications as directly observed therapy (DOT), and there is a risk of the women buying substandard or counterfeit drugs. Hence some providers were hesitant about prescribing it for purchase at commercial stores.

> *"The major obstacle is money and to walk there. . .the centre is not nearby."*
>
> *FGD-PW-FBF-S-01*

> *"We don't have the net to give to the pregnant women here. We only encourage them to try and buy. . .They should also call us for seminars on treating malaria in pregnancy. We also need instruments for the detection of malaria."*
>
> *SSI-FBF-S-01*

> *"with IPT, we run short, we ask them to buy in the market and that is costly for them. . . So, to go and tell them to buy, personally, I discourage it because they may even buy fake ones."*
>
> *SSI-PHC-U-03*

**Attitudes of ANC providers.** Participants also reported that the attitude of ANC providers at public health facilities discourages pregnant women from ANC attendance at public health facilities where they are more likely to freely access IPTp and ITNs. Pregnant women, caregivers, community leaders and TBAs cited instances where pregnant women experienced or shared their experiences about the negative attitudes of healthcare providers at public health facilities. Despite having limited access to IPTp and ITNs, pregnant women and caregivers reported that engaging with TBAs or faith-based birth attendants gave them positive experiences including acceptability, comfort and spiritual support. These were key reasons for their preference for TBA and faith-based birth homes.

*"One can say mummy [TBA] this is what is happening to me . . .they sit you down and explain things to you. . . the nurses [public hospital] are harsh . . .here, they play with us, sing, talk and advise us on what to do. . . it's not only during clinic day that you can ask questions, you are free to come anytime. . . in hospitals, it is not every one of the nurses that have time. . . you go and they shout at you like you are a child."*

*FGD-PW-TBA-R -02*

*"They [healthcare providers] shout at them. . .that one is common. . . they will just shout "sit down somewhere", and it is not meant to be so. . .because the women will not come. . . that's the place they get the net and medicine."*

*SSI-CG-PHC-R-04*

**Influence of familial and social networks.** Pregnant women frequently detailed the influence of relatives or caregivers such as mothers-in-law, sisters and husbands on their choice of ANC provider, which influences access to IPTp. Several caregivers reported a preference for TBA or faith-based birth attendants, which limits pregnant women's access to IPTp and ITNs. The few caregivers who indicated a preference for ANC services in public health facilities reported that their wards received ITNs and medicines to protect them from MiP although none mentioned IPTp specifically. Male caregivers further elaborated that they provide transport fees for ANC attendance and acquiring ITNs while female caregivers reported encouraging pregnant women to attend ANC services and reminders to take medications and use of ITNs.

*"My mother in law prefers to send me to a TBA, and she took me there."* FGD-PW-TBA-S-01
*"My husband didn't allow me to use a TBA. So I gave birth to one of my children in maternity house so that's why I have been going. . .my husband's sister told me that this place was good."*

*FGD-PW-PHC-S-02*

*"The clinic is near the house. . . I monitor her when it is time for antenatal and I give her money and send her to the hospital. . .I usually ask after her welfare and what was discussed there. . . If they give her drugs or they told her to go for scan, I make sure she does it."*

*SSI-CG-PHC-S-04*

**Concurrent use of multiple ANC providers.** Both caregivers and pregnant women frequently mentioned concurrent use of TBA and/or faith-based birth attendants with ANC services in public health facilities. Nearly all caregivers and some TBAs and faith-based birth attendants reported encouraging concurrent use of hospital ANC services, explaining that

different ANC providers address different needs such as spiritual, emotional and medical. They elaborated that they encouraged ANC attendance in public health facilities so that pregnant women will receive ITNs and medical tests. Some pregnant women confirmed this, indicating that during attendance at TBAs and/or faith-based birth homes, they were encouraged to access ANC services at public health facilities to access medical tests and medicines, most frequently, tetanus injections. TBAs who reported that they encouraged pregnant women to concurrently use their services and public health ANC services also indicated a cordial relationship with the public healthcare providers that they recommended.

*"I registered at my mother-in-law's TBA centre. . .My mother-in-law knows the people here at the health centre so she tells us that it not only her place that we should come but that we should also be going to the centre because they give me water or medicine."*

*FGD-PW-TBA-S01*

*"My pregnant women said they would not go there so I went with them to the hospital. . . they [public facility providers] give them drugs and injection so I make sure they go there too."*

*SSI-CG-FBF-R-01*

## Discussion

This study provides in-depth insights into the factors influencing ITN and IPT uptake at the individual, interpersonal, social and provider levels in lower socioeconomic communities within Ogun State, Nigeria. Table 2 provides a summary of the study findings between and within the various levels. While these findings show a high awareness of MiP issues and use of ITN in preventing MiP at all levels of the social structure, there is low awareness of IPTp at the individual and social levels and at the health system level, among TBAs and faith-based birth attendants. Although awareness of IPTp for MiP was high among public healthcare providers, some public healthcare providers reported the notion that IPTp is aimed at treating MiP rather than preventing MiP. The choice of ANC provider, which facilitates access to and utilisation of IPTp and ITN, is influenced by several factors including the experiences of relatives with ANC providers, attitudes of ANC providers and community perceptions of ANC providers. Concurrent use of multiple ANC providers and ANC providers' relationships present opportunities for synergy to broaden acceptability and coverage for IPTp and ITNs.

At the individual and social levels, there is high awareness about MiP and its consequences contrary to findings of a previous study conducted in Nasarawa and Cross River States in Nigeria [4]. Furthermore, the high awareness of ITN use as a preventive malarial strategy is similar to a recent study [17], but low awareness of IPTp at the individual and social levels, is in contrast with the findings of that previous study. Very few pregnant women could specify sulfadoxine/pyrimethamine or its brand names such as fancidar and amalar as the preventive antimalarial given at public health facilities. There was also poor knowledge of the dosage and frequency of IPTp at the individual, social and health levels, with some pregnant women misconceiving the injections that they received at ANC—possibly, tetanus injections—as IPTp. Given that nearly half of the pregnant women were recruited from public health centres, the low awareness of IPTp suggests inadequate health education at public health facilities, which contributes to the poor uptake of IPTp. It is plausible that the ANC group counselling at public health facilities provided inadequate details and/or was not reiterated during individual service provision due to insufficient numbers of healthcare providers grappling with large workloads.

**Table 2. Summary of study findings.**

| Participant | Level | Factors influencing IPTp uptake and use of ITNs |
|---|---|---|
| Pregnant women | *Individual* | High ITN awareness |
| | | Poor knowledge on how and when to use ITNs |
| | | Low IPTp awareness |
| | | Poor adherence to IPTp as DOT |
| | | Experienced and perceived medication side effects |
| | | Accessibility of ITNs and IPTp determined by the type of ANC provider visited |
| | | Fear of taking medications when they are not sick |
| | | Preference for TBAs and faith-based attendants ANC services |
| | | Use of vitamin supplements |
| | | Geographical accessibility |
| | | Financial barriers |
| Caregivers & community leaders | *Interpersonal and social* | Financial barriers |
| | | Little or no awareness of IPTp |
| | | High awareness and support for use of ITNs |
| | | Health provider attitudes at public health facilities |
| | | Encouraging concurrent use of various ANC providers |
| | | Stock-outs in public health facilities, purchase commercially |
| TBAs and faith-based providers | *Social and health system* | High awareness and support for ITN use |
| | | Little to no awareness of, or access to IPTp |
| | | Use of herbal remedies and spiritual water |
| | | Encouraging pregnant wards to use their services concurrently with public health ANC services |
| Health providers | *Health system* | High awareness and free access to ITNs and IPTp |
| | | Perception of SP as a treatment strategy for MiP |
| | | Routine stock-outs of ITNs and SP requiring prescriptions for access from commercial sources |
| | | Poor ANC attendance by pregnant women |
| | | Reluctance to prescribe SP for fear of it being obtained from unregulated sources |

Considering low literacy rates in the study area, healthcare providers in public health facilities should intensify IPTp sensitisation. They should also emphasise the importance of taking IPTp to prevent MiP during individual encounters with pregnant women at every ANC visit.

Similar to findings of studies conducted in Nigeria and other African countries [4, 18], at the individual level, the experience of perceived medication side effects or side effects from ITN use and pregnant women's hesitation to take medications during pregnancy in the absence of illness serve as barriers to improving uptake of IPTp. It is plausible that the experience of medication side effects further strengthens the perception of potential harm during pregnancy, thereby discouraging the uptake of IPTp when it is not provided as DOT. Added to that, combination-based medications such as haematinics or vitamin supplements may facilitate the perception that taking IPTp is superfluous.

Access to IPTp at the individual level is further influenced by other interpersonal and social factors, such as relatives and caregivers who influence the type of ANC providers that pregnant women opt to access. As such, our findings regarding a preference for TBA ANC services at the individual and social levels are consistent with existing literature [9, 10]. The preference for TBAs appears informed by the dignity and respect that pregnant women and caregivers

feel that they receive from TBAs. This suggests, similar to findings in other studies [19–21], that poor quality of care at public health facilities remains a key barrier to improving pregnant women's utilisation of public health facilities. While ANC providers' poor attitudes and interpersonal skills associated with poor quality of care provided were stated as key reasons for the preference and trust in TBA ANC services, there was added social support for concurrent use of both or all types of ANC providers in this study. This perhaps reflects the realities of integrating traditional healthcare practices and biomedical healthcare as well as changing health-seeking behaviours. It plausibly reflects a health-seeking behaviour to get the "best of both worlds" of healthcare by accessing traditional behavioural maternal support and the benefits of biomedical health care for pregnancy outcomes. While this provides an opportunity to further enhance the acceptability and uptake of IPTp among attending pregnant women, this health-seeking behaviour could further be harnessed by building the capacities of health care provider alternatives regarding MiP and the value of IPTp.

While financial and geographical barriers to ANC services in public health facilities cut across the individual and social levels, their impact on access to ITNs and IPTp remain prevalent as shown in this and other studies [6, 17, 18]. Costs associated with procurement of SP and ITNs when they are out of stock at public health facilities directly affect pregnant women's access to IPTp and ITNs. Caregivers and family members (most often male partners) have the added burden of providing the finances needed to procure these services and also, due to long distances to public health facilities, have to provide for the transportation and upkeep of pregnant women. While non-governmental organisations assist to improve free access to ITNs in rural and economically disadvantaged communities, more commitment and synergy is needed between civil organisations and governments to ensure that such stock-outs do not create financial burdens that entrench poverty within such settings and further hinder the uptake of IPTp.

In line with health policy directives at the health system level, IPTp is expected to be available at both public and private ANC facilities [4] yet, this study shows poor awareness of IPTp among ANC providers—faith-based birthing homes and TBAs (aspects of the health system), among caregivers and community leaders (social level) and pregnant women (individual). While IPTp awareness was high among ANC providers in public health facilities, the poor IPTp awareness among faith-based birth attendants and TBAs has implications for the awareness, acceptability, access and uptake of IPTp. With about 60% of Nigerians accessing health care from private health facilities, and TBAs providing, a significant amount of ANC services [7, 9, 10]; the poor awareness of IPTp by these providers hinders access to IPTp, which they are unlikely to provide as part of their ANC services. The few TBAs and faith-based providers with good relationships with public health service ANC providers encouraged attending pregnant women to concurrently access ANC services at public health facilities. This potentially reflects TBAs' recognition of public health service ANC providers' knowledge and expertise, which highlights the synergy that could be harnessed to expand IPTp uptake and contribute to improved pregnancy outcomes. Health care policy planners and implementers including the government should leverage and develop approaches to ensure delivery of MiP control programmes by healthcare providers in the private sector. Given their wider reach to pregnant women, this would improve the acceptability and uptake of IPTp by encouraging referrals to public health facilities. Indeed, studies have shown the potential that incorporating TBAs could reduce adverse maternal and child health outcomes in various settings [22, 23].

Similar to findings in other studies [17, 24], drug stock-outs in public health facilities contribute to the low coverage of IPTp, highlighting lapses in the drug supply chain at all levels and the need for improved efficiency. It potentially explains why the NDHS 2018 found that despite over 57% of pregnant women attending at least four ANC visits in the last two years,

only 17% of them with a live birth had received at least three IPTp doses [5]. Such drug stock-outs require that ANC providers prescribe IPTp for pregnant women to buy from commercial sources, which places financial constraints on pregnant women and their families. Also, healthcare providers are reluctant to prescribe IPTp for pregnant women to access from commercial sources for fear of counterfeit medications via unregulated drug outlets. This confirms findings at the individual and social levels where financial constraints to IPTp access were reported due to stock-outs at the public health facility. Non-adherence to the national policy recommendation of administering IPTp as a directly observed therapy [25], also appears to be a key barrier to the patchy implementation of IPTp in this region. Given the concerns regarding poor quality drugs, particularly in the African region [26], more effort is needed to identify and regulate community-based drug outlets while creating awareness among ANC providers on regulated drug outlets.

## Conclusion

The study highlights a crucial need for health literacy programs that underscore the importance of accessing IPTp at all levels of the social structure while addressing the social and health sector factors contributing to the poor uptake of IPTp. In particular, opportunities exist for leveraging on changing health-seeking behaviours at the individual and social levels. Health implementers should network with TBAs and faith-based ANC providers to improve pregnant women's access to and utilisation of IPTp towards preventing MiP. Health literacy programs on MiP should extend beyond pregnant women, families and societies to TBAs and faith-based ANC providers. Public health facility providers should be encouraged to provide respectful, empathic and dignified care to improve uptake of evidenced effective MiP interventions. While other health sector barriers to IPTp need to be addressed, there is a need for public health officials to engage and educate TBAs and faith-based providers on MiP and IPTp towards the goal of improving maternal and new-born health outcomes.

## Supporting information

**S1 File.**
(DOCX)

**S2 File.**
(PDF)

## Acknowledgments

The authors also thank the leaders of the study communities and the study participants for their participation. We also appreciate the directors and staff of the ANC facilities who assisted in the identification and recruitment of the study participants.

## Author Contributions

**Conceptualization:** Gertrude N. Nyaaba, Atinuke O. Olaleye, Mary O. Obiyan, Dilly O. C. Anumba.

**Formal analysis:** Gertrude N. Nyaaba.

**Funding acquisition:** Atinuke O. Olaleye, Oladapo Walker, Dilly O. C. Anumba.

**Methodology:** Gertrude N. Nyaaba, Atinuke O. Olaleye, Mary O. Obiyan.

**Project administration:** Atinuke O. Olaleye, Mary O. Obiyan, Dilly O. C. Anumba.

**Resources:** Oladapo Walker, Dilly O. C. Anumba.

**Supervision:** Dilly O. C. Anumba.

**Validation:** Mary O. Obiyan.

**Visualization:** Gertrude N. Nyaaba.

**Writing – original draft:** Gertrude N. Nyaaba.

**Writing – review & editing:** Gertrude N. Nyaaba, Atinuke O. Olaleye, Mary O. Obiyan, Oladapo Walker, Dilly O. C. Anumba.

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
