## [Decision Letter · Decision Letter 0]

22 Oct 2020

PONE-D-20-28916

A socio-ecological approach to understanding the factors influencing the uptake of intermittent preventive treatment of malaria in pregnancy (IPTp) in South Western Nigeria.

PLOS ONE

Dear Dr. Nyaaba,

Thank you for submitting your manuscript to PLOS ONE. After careful consideration, we feel that it has merit but does not fully meet PLOS ONE’s publication criteria as it currently stands. Therefore, we invite you to submit a revised version of the manuscript that addresses the points raised during the review process.Please submit your revised manuscript by Dec 06 2020 11:59PM. If you will need more time than this to complete your revisions, please reply to this message or contact the journal office at plosone@plos.org. Please include the following items when submitting your revised manuscript:

We look forward to receiving your revised manuscript.

Kind regards,

Khin Thet Wai, MBBS, MPH, MA (Population & Family Planning Resear

Academic Editor

PLOS ONE

Journal Requirements:

"We thank the Research England, University of Sheffield Sustainable partnerships Grant body for funding this project. We acknowledge the support of the National Institute for Health Research Global Health Research Group on Preterm Birth Prevention and Management (NIHR PRIME). [...]"

"This manuscript is part of a larger study funded by a QR-GCRF (Research England, University of Sheffield) Sustainable partnerships Grant X/159770 awarded to Professor Dilly OC Anumba. The funders did not play any role in the design of the study, data collection, analysis and development of this manuscript."

Reviewers' comments:

Reviewer's Responses to Questions

**Comments to the Author**

1. Is the manuscript technically sound, and do the data support the conclusions?

Reviewer #1: Yes

Reviewer #2: Yes

2. Has the statistical analysis been performed appropriately and rigorously? 

Reviewer #1: N/A

Reviewer #2: N/A

3. Have the authors made all data underlying the findings in their manuscript fully available?

Reviewer #1: Yes

Reviewer #2: Yes

4. Is the manuscript presented in an intelligible fashion and written in standard English?

Reviewer #1: No

Reviewer #2: Yes

5. Review Comments to the Author

Reviewer #1: Reviewer comment for PLOS ONE

Date of review: 10 Oct 2020

Manuscript Number: PONE-D-20-28916

Article Type: Research Article

Full Title: A socio-ecological approach to understanding the factors influencing the uptake of intermittent preventive treatment of malaria in pregnancy (IPTp) in South Western Nigeria.

Corresponding Author: Gertrude Nsorma Nyaaba, PhD, The University of Sheffield, Sheffield, South Yorkshire UNITED KINGDOM

Reviewer comment

Summary of the research and overall impression

This manuscript identify access to and utilization of IPT and ITN among pregnant women for malaria prevention and control from the different perceptive of pregnant women, local people, different service providers and health facilities. The findings will be useful for national program to understand the evidence-based policy recommendation in the area for improvement for better access to and utilization of IPT and ITN among pregnant women.

Generally, the introduction and methods sessions are well presented. English langue editing is suggested, especially in the results (findings) which needed to be more concrete and shorter with the scientific writing. The language in quotations should also be edited to be more appropriate in layman term description.

Discussion on specific area of improvement

Abstract:

Well-written and clearly presented abstract. A few minor English language editing is required.

1.Eg. Line 23. “This study explores…” it should be with past tense – “explored”. Past tense should be used for results session and conclusion (the last paragraph).

2.In last sentence, two words are duplicating “improving… xxx …..improving….xxx”. Please rewrite.

Introduction

Generally, the information in the introduction is comprehensive, a few minor English language editing is required and some references should be updated.

3.Line 47-48: please update the reference. The WHO world malaria report 2019 is available for updated data.

4.Line 64-67: the sentence is not clear in writing. Please rewrite.

5.Objectives of the study is missing in the introduction. Please add the objectives at the end of the introduction.

Materials and methods

Generally, comprehensive and clearly stated the study design, methods and data analysis. A few minor comments below.

6.Methods: is the sample size for SSI and FGD were pre-defined or how the samples size was determined? Did you continue the interviews until the information (or data) was saturated? Please add.

7.Data analysis: please mention data analysis method (eg. “thematic approach” as stated in the abstract.)

Results

8.Table 1: A. FGD participants is suggested to change as followed.

Location Place of ANC attending # of FGD conducted# of participants per FGDAge range # of children Education Language

Rural location Primary health center

TBA center

Faith-based center

Semi-urban location Primary health center

TBA center

Faith-based center

Urban location Primary health center

Faith-based center

9.“ID” column is suggested to delete from both table A and B. Please delete “duration” column from table B.

10.In qualitative data, quotations should be carefully re-analyzed.

a.The number of quotations must be reduced. Recommend to select ONE best suitable quotation that reflect to the finding description is good enough (a maximum of two, not more than two).

b.Participant’s ID number [eg. FGD-PW-TBA-S-02] must be removed from the selected quotation. Instead, please describe [eg. a 28-year pregnant woman from FGD who took ANC at TBA clinic (OR) a pregnant woman from FGD who took ANC at TBA clinic]

c.Some quotations are related to mis-believe, please describe about it in the description above, what kind of mis-believe they have. (eg. line 150-156)

11.Line 145-146: please mention what are the “personal and social network experiences”.

12.Line 163: the descriptions are more related to their local practices for prevention of malaria. Suggested to change the sub-title “Practices on malaria preventive measures”, instead of “Preventing MiP”.

13.Line 204-222: the title of the description is “Access to and utilization of IPT”, but the paragraph is mixed information of about IPT and other local traditional medicine or mis-believe practices as the pre-medication of malaria prevention. Please keep the description about access to and utilization of IPT, and separately mention in the next paragraph for “other pre-medication practices”. Thus, please re-arrange the description and quotations for line 204-233.

14.“Access to and utilization of ITN” and “Access to and utilization of IPT” should be first level title, not the second level sub-title under the “Malaria in pregnancy”.

15.Line 224: Overall, I would suggest the title as “Reasons for poor IPT uptake and less use of ITN” instead of “Factors influencing IPT uptake and use of ITN”. This session is mixed different findings and better to have separate sub-titles. Edit the English language in the descriptions and reduce the writing to be more concrete and scientific writing. Please re-organize the findings (descriptions and quotations) with the following sub-titles.

a.Fear of side-effect (line 235-239)

b.Lack of knowledge and poor health education (line 249-257)

What do you mean by “sensitisation” in line 249? Please clarify or use another suitable terminology.

c.Financial barriers (line 267-276)

d.Attitude of health care providers (line 290-299)

e.Influence of decision makers (line 315-323)

f.Public health facilities versus TBA or faith-based clinics (line 338-349)

Discussion and conclusions

16.General comment

a.Discussion should be re-written for more comprehensiveness, point by point. You may want to describe as “First”, “Second”, “Third” points, etc. And discussion should lead to the recommendations for the public health importance and policy input.

b.The term “in line with the literature” should not be used and please elaborate what are the similar findings or difference in findings from other literatures.

c.Please cite more references for the findings and recommendation in the discussion session.

d.Please make sure the point in the discussion must be consistent with and mentioned in the findings. Eg. findings about drug stock out was not clearly mentioned in the results session, but the discussion talked about the drug stock out for a paragraph. Please re-check.

e.Conclusions should be linked to the discussion and focus on the recommendation and policy implication from the study findings.

Other points

Recommendation: I would like to propose the major revision for this manuscript.

Reviewer #2: Review comments-PONE-D-20-28916

Minor concern:

Abstract:

Line 2: Insert ‘in’ between particularly and Africa.

Results:

Table 1: row 1; column 4 “No participants” using ‘No’ to represent number, may be misleading.

Major concern:

Materials and methods:

Provide a description of the study site/area.

6. PLOS authors have the option to publish the peer review history of their article (what does this mean?). If published, this will include your full peer review and any attached files.

Reviewer #1: **Yes: **Poe Poe Aung

Reviewer #2: **Yes: **Prof. Francis Anto

---

## [Author Response · Author response to Decision Letter 0]

16 Jan 2021

Response to reviewers comments attached

---

## [Decision Letter · Decision Letter 1]

26 Feb 2021

A socio-ecological approach to understanding the factors influencing the uptake of intermittent preventive treatment of malaria in pregnancy (IPTp) in South Western Nigeria.

PONE-D-20-28916R1

Dear Dr. Nyaaba,

We’re pleased to inform you that your manuscript has been judged scientifically suitable for publication and will be formally accepted for publication once it meets all outstanding technical requirements.

Kind regards,

Khin Thet Wai, MBBS, MPH, MA (Population & Family Planning Res.)

Academic Editor

PLOS ONE

Additional Editor Comments (optional):

Reviewers' comments:

Reviewer's Responses to Questions

**Comments to the Author**

1. If the authors have adequately addressed your comments raised in a previous round of review and you feel that this manuscript is now acceptable for publication, you may indicate that here to bypass the “Comments to the Author” section, enter your conflict of interest statement in the “Confidential to Editor” section, and submit your "Accept" recommendation.

Reviewer #1: (No Response)

Reviewer #2: All comments have been addressed

2. Is the manuscript technically sound, and do the data support the conclusions?

Reviewer #1: Yes

Reviewer #2: Yes

3. Has the statistical analysis been performed appropriately and rigorously? 

Reviewer #1: N/A

Reviewer #2: N/A

4. Have the authors made all data underlying the findings in their manuscript fully available?

Reviewer #1: No

Reviewer #2: Yes

5. Is the manuscript presented in an intelligible fashion and written in standard English?

Reviewer #1: Yes

Reviewer #2: Yes

6. Review Comments to the Author

Reviewer #1: (No Response)

Reviewer #2: The authors have adequately addressed all the concerns (including a detailed description of the study site) raised in the original review..

7. PLOS authors have the option to publish the peer review history of their article (what does this mean?). If published, this will include your full peer review and any attached files.

Reviewer #1: **Yes: **Poe Poe Aung

Reviewer #2: **Yes: **Francis Anto

---

## [Editor Report · Acceptance letter]

4 Mar 2021

PONE-D-20-28916R1 

A socio-ecological approach to understanding the factors influencing the uptake of intermittent preventive treatment of malaria in pregnancy (IPTp) in South-Western Nigeria. 

Dear Dr. Nyaaba:

I'm pleased to inform you that your manuscript has been deemed suitable for publication in PLOS ONE. Congratulations! Your manuscript is now with our production department. 

Kind regards, 

on behalf of

Dr. Khin Thet Wai 

Academic Editor

PLOS ONE